# Association of Health Disparities with Glioblastoma Treatment and Outcomes: Insights from a 15-Year National Cohort (2005–2020) [note 1]

**DOI:** 10.3390/brainsci15060556

**Published:** 2025-05-23

**Authors:** Zouina Sarfraz, Diya Jayram, Ahmad Ozair, Lydia Hodgson, Shreyas Bellur, Arun Maharaj, Alireza Mansouri, Manmeet S. Ahluwalia

**Affiliations:** 1Miami Cancer Institute, Baptist Health South Florida, Miami, FL 33176, USA; zouina.sarfraz@baptisthealth.net (Z.S.); diya.jayram@baptisthealth.net (D.J.); ahmad.ozair@baptisthealth.net (A.O.);; 2Department of Neurosurgery, University of Maryland, Baltimore, MD 20742, USA; 3Penn State Cancer Institute, Penn State Health Milton S. Hershey Medical Center, Hershey, PA 17033, USA; 4Department of Neurosurgery, Penn State Health Milton S. Hershey Medical Center, Hershey, PA 17033, USA; 5Herbert Wertheim College of Medicine, Florida International University, Miami, FL 33199, USA

**Keywords:** high-grade glioma, neuro-oncology, racial disparities, molecular profiling, healthcare inequity, astrocytoma, brain cancer

## Abstract

**Background:** Despite advances in glioblastoma (GBM) management, median overall survival (mOS) remains poor, and multi-modal disparities persist. We sought to evaluate trends in GBM treatment and survival outcomes from 2005–2020, with a focus on sociodemographic and geographic disparities. **Methods:** We conducted a retrospective US-based cohort study using the National Cancer Database (NCDB), stratifying study period into four intervals (2005–2008, 2009–2012, 2013–2016, and 2017–2020). Logistic regression was used to identified predictors of receipt of combination surgery, radiation, and chemotherapy (Sx+RT+Chemo). Kaplan–Meier and multivariable Cox proportional hazards approaches were used to assess mOS. **Results:** A total of 111,955 adults with GBM were included. From 2005–2008 to 2017–2020, mOS increased from 7.8 to 9.5 months, with geographically unequal gains in survival across the US. In multivariable logistic regression model adjusting for known confounders, combined Sx+RT+Chemo was less likely to be received by female patients (OR 0.90, 95% CI 0.88–0.92) vs. male, non-White patients (OR 0.90, 95% CI 0.86–0.94) vs. White, patients treated at community hospitals (OR: 0.78, 95% CI 0.76–0.80) vs. academic centers, publicly-insured patients (OR 0.74, 95% CI 0.71–0.76) or uninsured patients (OR 0.54, 95% CI 0.50–0.58) vs. privately-insured, and patients living in the South (OR 0.88, 95% CI 0.85–0.91), Midwest (OR 0.83, 95% CI 0.80–0.86), and West (OR 0.85, 95% CI 0.81–0.88) compared to the Northeast. In multivariable Cox regression, significantly poorer survival was seen amongst non-metropolitan patients, community-based hospital patients, and publicly-insured and uninsured patients (vs. privately-insured), despite adjusting for prognostic factors. **Conclusions:** Only modest improvement in mOS of GBM patients has occurred across 2005–2020, with persistent disparities linked to sociodemographic and structural factors, whose redressal warrants multi-pronged efforts.

## 1. Introduction

Glioblastoma (GBM) is the most common primary brain tumor in adults and accounts for 45.2% of all primary malignant brain and central nervous system (CNS) tumors [1,2]. As an aggressive and grade 4 astrocytic tumor, it is characterized by diffuse infiltration, necrosis and microvascular proliferation [3,4]. Therapeutic advances in GBM remain challenging given the rapid-growing, invasive nature of the disease and blood–brain barrier penetration limitations, which hinders effective drug delivery [5,6]. Conventional treatment modalities include maximal safe surgical resection followed by radiation and chemotherapy [7,8]. Recent approaches include tumor treating fields, targeted brachytherapy, and molecularly guided therapy, which are now being adopted in specialized settings [9,10]. Despite these advances, the prognosis of GBM remains poor, with a 5-year survival rate of 5.6% in adults aged ≥40 years [11,12]. These persistent sobering survival outcomes have been, accompanied by advances in our understanding of tumor biology and classification [13,14]. Molecular profiling with features such as O6-methylguanine-DNA methyltransferase (MGMT) promoter methylation, observed in 35–40% of GBM cases, predicts improved responses to alkylating chemotherapy, associated with better survival outcomes [15,16,17]. However, despite advances in molecular understanding and treatment of GBM, there remains a paucity of large-scale, population-based data evaluating how sociodemographic disparities influence access to care and outcomes. This study aims to evaluate the trends in GBM management strategies and clinical outcomes in a large and diverse patient population across the United States (US), as well as to examine disparities in care and survival.

## 2. Materials and Methods

### 2.1. Study Design and Data Source

This was a retrospective cohort study performed using the National Cancer Database (NCDB), a cancer dataset jointly maintained by the Commission on Cancer of the American College of Surgeons and the American Cancer Society. The work is reported following the STROBE reporting guidelines (STROBE Flow Diagram in Figure 1). The NCDB contains hospital registry data from over 1500 accredited facilities that represent nearly 70% of newly diagnosed cancers cases across the country [18]. Data are abstracted by certified tumor registrars at each participating institution and submitted following standardized coding protocols to ensure uniformity and quality across the dataset. National trends in GBM demographics, treatment patterns, and survival outcomes were analyzed. To assess temporal shifts, the 15-year study period was divided into four approximately equal intervals (2005–2008, 2009–2012, 2013–2016, and 2017–2020), allowing for evaluation of evolving treatment practices and disparities across consistent timeframes. The intervals were chosen to balance temporal resolution with sufficient sample sizes for robust statistical comparisons. The preliminary findings of this study were presented prior [19]. This project is registered in the Open Science Framework: DOI 10.17605/OSF.IO/VJ6FQ.

### 2.2. Study Population

Patients were diagnosed with primary GBM prior to study enrollment. They were identified using the International Classification of Diseases for Oncology—Third Edition (ICD-O-3) morphology codes [20], which are further delineated in Appendix A. For consistency and to assess temporal trends, the data were stratified into four periods: 2005–2008, 2009–2012, 2013–2016, and 2017–2020. Patients with a survival time of 0 days or unknown laterality were excluded to ensure the completeness and reliability of the final analytic cohort (Figure 1).

### 2.3. Variables and Measures

Patient demographics included age at diagnosis (categorized as 40–49, 50–64, 65–74, and 74–90 years), sex (male, female), race (White, non-White), ethnicity (Hispanic, non-Hispanic), geographic region (Northeast, South, Midwest, West), household income, and facility type (academic, community, or network). Median household income was estimated using the American Community Survey (ACS) 2016–2020 5-year estimates, as matched by patient ZIP code. Income was categorized as <$74,063 versus ≥$74,063, based on the NCDB 2022 Participant User File income quartiles. The income cutoff of $74,063 was selected to align closely with the national median household income reported by the U.S. Census Bureau for 2022. According to the 2023 Current Population Survey Annual Social and Economic Supplement (CPS ASEC), the real median household income in the United States was $74,580 in 2022 [21]. Given that the patient cohort spans diagnoses from 2005 to 2020, $74,063 represents a conservative threshold reflective of contemporary national medians while accounting for inflation adjustments and ensuring consistency across socioeconomic analyses. Educational attainment was defined by the percentage of adults without a high school diploma in the patient’s ZIP code, derived from the ACS 2016–2020 data, and categorized as ≥9.1% vs. <9.1%. Urbanicity was determined using the NCDB-provided urban/rural classification based on U.S. Department of Agriculture urban influence codes (metro vs. non-metro).

Clinical variables included primary tumor site, tumor laterality (unilateral vs. bilateral/midline), histologic subtype (glioblastoma NOS, gliosarcoma, giant cell glioblastoma, and IDH-mutant glioblastoma), and comorbidity burden assessed by the Charlson–Deyo Comorbidity Index (0, 1, 2, 3). Treatment characteristics included receipt of surgery, radiation therapy, chemotherapy (Sx+RT+Chemo), and immunotherapy, as well as facility type and time interval between diagnosis and initiation of radiation. Outcomes evaluated were overall survival (OS), 30-day mortality, 90-day mortality, and 30-day hospital readmission rates.

### 2.4. Statistical Analysis

Descriptive statistics were used to summarize patient demographic, clinical, and treatment characteristics. Continuous variables were reported as means with standard deviations (SD) or medians with interquartile ranges (IQRs), depending on distribution, and categorical variables were summarized as frequencies and percentages. Comparisons across the four study periods (2005–2008, 2009–2012, 2013–2016, and 2017–2020) were performed using Chi-square tests for categorical variables and one-way analysis of variance (ANOVA) or Kruskal–Wallis tests for continuous variables, based on normality assessments.

Overall survival (OS) was estimated using Kaplan–Meier methods, and survival differences between groups were evaluated using the log-rank test. Multivariable Cox proportional hazards regression models were fitted to identify independent predictors of OS, adjusting for demographic, clinical, and treatment-related covariates. The proportional hazards assumption was assessed using scaled Schoenfeld residuals. Minor deviations from proportionality were noted for selected covariates (e.g., insurance status, urbanicity, treatment type, diagnosis year, and age). Multicollinearity was assessed using variance inflation factors (VIFs), with all included variables demonstrating VIFs < 2. To address these deviations and validate findings, supplementary parametric survival analyses were conducted using accelerated failure time (AFT) models under Weibull, log-normal, and log-logistic distributions. The AFT framework was selected because it models survival times directly rather than hazard rates, offering a robust alternative when the proportional hazards assumption is not fully satisfied.

Multivariable logistic regression analyses were also performed to identify factors associated with the receipt of combined Sx+RT+Chemo.

Complete case analysis was applied for variables with missing data. All analyses were conducted using RStudio (version 4.1.2; The R Foundation for Statistical Computing). A two-sided *p*-value of ≤0.05 was considered statistically significant. No formal adjustment for multiple comparisons was performed; results from secondary and exploratory analyses should be interpreted with caution due to the potential for Type I error inflation.

## 3. Results

### 3.1. Cohort Characteristics

In total, 111,955 patients between 2005 and 2020 were included in this analysis (Table 1). Of these, 8.9% of patients were aged 40–49 years, 38.4% were 50–64 years, 30.1% were 65–74 years, and 22.7% were 74–90 years at diagnosis. Males comprised 57.5% of the cohort, and females accounted for 42.5%. The majority of patients were White (91.3%), while Non-White patients represented 8.7%. Regarding ethnicity, 94.7% of patients were non-Hispanic and 5.3% were Hispanic. Treatment facilities included academic centers (41.3%), community hospitals (39.0%), and network facilities (19.7%). A proportion of 59.8% of patients resided in areas with a median household income below $74,063, while 40.2% lived in areas with income at or above $74,063. Educational attainment, defined by the proportion of adults aged 25 years or older in the ZIP code without a high school diploma, showed that 44.8% of patients lived in areas where ≥9.1% of adults had not graduated, whereas 55.2% lived in areas with lower rates of non-completion. Most patients (84.1%) resided in metropolitan areas, while 15.9% lived in non-metropolitan regions. Regarding comorbidity, Charlson–Deyo scores were distributed as 69.6% with a score of 0, 17.5% with a score of 1, 8.0% with a score of 2, and 4.9% with a score of 3. Insurance status included private insurance (41.0%), government insurance (55.9%), and uninsured (3.1%). Geographically, 36.8% of patients lived in the Northeast, 28.5% in the South, 20.3% in the Midwest, and 14.4% in the West. Histologically, 96.8% of tumors were classified as GBM, 2.1% as gliosarcoma, 0.7% as giant cell GBM, and 0.3% as IDH-mutant GBM. Tumor laterality was unilateral in 74.0% of cases and bilateral or midline in 26.0%. Primary tumor sites included the frontal-temporal lobes (52.1%), parietal-occipital lobes (19.6%), overlapping lesions (14.2%), ventricles or cerebellum (0.9%), brainstem (0.3%), brain not otherwise specified (9.1%), and cerebrum (3.7%) (Table 1).

### 3.2. Temporal Trends and Treatment Characteristics

Temporal trends in treatment delivery and early outcomes are summarized in Table 2. Use of chemotherapy modestly increased over the study period, from 62.8% in 2005–2008 to 68.3% in 2017–2020. Radiation therapy and surgical treatment also saw slight increases in use, from 69.7% to 72.2% and from 72.3% to 77.2%, respectively. Immunotherapy use remained limited overall but was more frequently recorded in recent years, increasing from 0.3% to 6.3% across the time periods.

The median time from diagnosis to the start of radiation increased from 29.0 to 36.0 days. Meanwhile, the median number of elapsed days for radiation therapy declined from 42.0 to 37.0 days. Median length of hospital stay following surgical treatment decreased from 4.0 days to 3.0 days.

Early outcomes remained generally stable or showed modest changes. Thirty-day mortality decreased from 6.2% to 4.3%, and ninety-day mortality from 18.0% to 14.5% over the study period. Thirty-day hospital readmission rates remained between 5.6% and 5.2%. Median overall survival increased slightly from 7.8 months in 2005–2008 to 9.5 months in 2017–2020.

### 3.3. Predictors of Receipt of Combined Surgery + Radiation + Chemotherapy

In multivariable logistic regression, several demographic, clinical, and socioeconomic factors were associated with a higher likelihood of receiving combined surgery, radiation, and chemotherapy (Sx+RT+Chemo) (Figure 2). Patients aged ≥65 years (OR: 0.52, 95% CI: 0.50–0.53), female patients (OR: 0.90, 95% CI: 0.88–0.92), and non-White patients (OR: 0.90, 95% CI: 0.86–0.94) had lower odds of receiving combined Sx+RT+Chemo. Treatment at community (OR: 0.78, 95% CI: 0.76–0.80) or network facilities (OR: 0.87, 95% CI: 0.84–0.90) was also associated with lower odds compared to academic centers. Patients residing in areas with higher median income (≥$74,063) (OR: 1.15, 95% CI: 1.12–1.19) and lower proportions of adults without high school education (<9.1%) (OR: 1.14, 95% CI: 1.11–1.17) were more likely to receive combined Sx+RT+Chemo.

Higher comorbidity burden was associated with reduced odds of treatment, with a stepwise decline from Charlson–Deyo scores of 1 (OR: 0.86, 95% CI: 0.83–0.89) to 3 (OR: 0.62, 95% CI: 0.59–0.66). Compared to patients with private insurance, those with government insurance (OR: 0.74, 95% CI: 0.71–0.76) and uninsured patients (OR: 0.54, 95% CI: 0.50–0.58) had significantly lower odds of receiving combined Sx+RT+Chemo. Regional variation was also observed, with lower odds of combination therapy receipt in the South (OR: 0.88, 95% CI: 0.85–0.91), the Midwest (OR: 0.83, 95% CI: 0.80–0.86), and the West (OR: 0.85, 95% CI: 0.81–0.88) compared to the Northeast US.

A visual summary of the adjusted odds ratios is presented in Figure 2. Full model and reduced coefficients are provided in Appendix A, respectively.

### 3.4. Overall Survival by Treatment Category

For the entire cohort, the median OS was 9.30 months (95% CI: 9.20–9.40). Patients receiving combined Sx+RT+Chemo had a median OS of 14.62 months (95% CI: 14.52–14.75), with 1-year and 3-year survival rates of 59.9% and 15.2%, respectively. Patients receiving RT+Chemo or Sx+RT had median OS of 7.00 months (95% CI: 6.80–7.20) and 7.16 months (95% CI: 6.90–7.39), respectively. Median OS among patients receiving surgery alone or other therapies was 3.09 months (95% CI: 3.00–3.15) and 4.57 months (95% CI: 4.44–4.73), respectively. Patients who did not receive surgery, radiation, or chemotherapy had a median OS of 1.64 months (95% CI: 1.61–1.68) (Table 3, Figure 3).

### 3.5. Overall Survival by Receipt of Combined Surgery, Radiation, and Chemotherapy

Among all patients, the median OS was 9.30 months (95% CI: 9.20–9.40). Patients receiving combined Sx+RT+Chemo had a median OS of 14.62 months (95% CI: 14.52–14.75), whereas patients not receiving combination therapy had a median OS of 3.84 months (95% CI: 3.81–3.91). At 1 year, the survival rate was 59.9% among combination therapy recipients compared to 21.0% among non-recipients. The 3-year survival rates were 15.2% and 5.8%, respectively (Table 4, Figure 4).

### 3.6. Predictors of Overall Survival (Cox Models)

Multivariable Cox proportional hazards regression findings are summarized in Table 5.

Older age (≥65 years) was associated with poorer survival (aHR 1.43, 95% CI 1.41–1.46), while female sex was modestly associated with improved survival (aHR 0.95, 95% CI 0.94–0.96). Non-White race (aHR 0.78, 95% CI 0.76–0.80) and Hispanic ethnicity (aHR 0.75, 95% CI 0.73–0.77) were associated with better survival compared with White and non-Hispanic patients, respectively.

Treatment at community and network facilities was associated with poorer survival compared with academic centers (aHRs 1.09 and 1.12, respectively). Additional factors associated with inferior survival included lower household income, higher Charlson–Deyo comorbidity scores (≥1), residence in non-metropolitan areas, and bilateral or midline tumor laterality.

Scaled Schoenfeld residuals were used to evaluate the proportional hazards assumption (Appendix A). Residuals for race, laterality, and geographic region were generally flat, supporting proportionality. Minor deviations were observed for insurance status, urbanicity, treatment type, age, and diagnosis year, but these were modest and not deemed clinically meaningful given the large cohort. Overall, the Cox proportional hazards model was considered appropriate.

Sensitivity analyses using stratified Cox models by histology, facility type, age, insurance, comorbidity burden, and geographic region demonstrated consistent findings and are presented in Appendix A.

### 3.7. Sensitivity Analyses Using Parametric Survival Models

Given minor deviations from proportional hazards, particularly for insurance, urbanicity, treatment, age, and year, we supplemented our analysis with parametric AFT models. AFT models offer a robust alternative when proportionality is imperfect, and findings were broadly consistent with Cox results, with time ratios (TRs) < 1 indicating shorter survival and TRs >1 indicating longer survival.

Predictors of interest of shorter survival included older age (≥65 years; TR 0.67, 95% CI 0.66–0.68), treatment at community or network facilities compared to academic centers, lower income, higher comorbidity burden, residence in non-metropolitan areas, and bilateral or midline tumor laterality. In contrast, female sex (TR 1.07, 95% CI 1.06–1.08), Non-White race (TR 1.33, 95% CI 1.30–1.36), and Hispanic ethnicity (TR 1.40, 95% CI 1.34–1.46) were associated with longer survival.

Results from the Weibull AFT model are presented in Table 6. Appendix A present findings from log-normal and log-logistic AFT models, respectively, which yielded comparable results.

## 4. Discussion

This study evaluated trends, progress, and ongoing challenges in GBM management across a 15-year period. In patients receiving combined Sx+RT+Chemo the median OS was 14.6 months. These findings are consistent with prior reports, such as that of Delgado-López et al., who documented a median OS of 14 months following maximum safe resection and chemoradiotherapy [22]. The use of triple-modality therapy increased by approximately 8% from 2005 to 2020, reflecting improved adoption of evidence-based standards of care. Despite these gains, demographic, socioeconomic, and geographic disparities continued to impact treatment receipt and survival outcomes. These results align with broader national trends toward optimizing GBM care [23,24,25].

Thomas-Joulié and colleagues reported that median OS improved between the 2005–2012 and 2013–2018 periods, with a HR of 0.64 (*p* < 0.001) favoring the more recent cohort [26]. These findings were attributed to advancements in supportive care, earlier GBM detection, and optimization of treatment delivery. Similarly, Touat et al. highlighted that modest survival gains may also reflect greater adoption of molecular markers, such as MGMT promoter methylation, for treatment stratification [26,27,28]. Despite these developments, median survival for GBM remains limited at 14–20 months, and the 5-year survival rate remains approximately 5%, with prognosis heavily influenced by factors, such as patient age, performance status, molecular characteristics, and extent of surgical resection [29,30]. The aggressive, infiltrative nature of GBM further compounds the challenges related to systemic therapy penetration across the blood–brain barrier [31].

While this study observed increasing receipt of combined Sx+RT+Chemo over time, improvements in perioperative and postoperative care also contributed to significantly reduced early mortality. The decline in 30- and 90-day mortality rates across the study period suggests improved surgical techniques, better management of comorbidities, and reductions in treatment-related toxicities, all of which are pivotal components of modern GBM care [32,33]. Supportive care interventions remain critical in minimizing side effects and optimizing outcomes in glioblastoma management [26]. Nonetheless, these advances remain modest. Despite improvements in supportive care and multimodal treatment delivery, the “Stupp protocol”, introduced in the mid-2000s, continues to define the standard of care for GBM [33,34]. Emerging targeted therapies and immunotherapies have yet to demonstrate consistent benefit in clinical trials, largely due to the immunosuppressive tumor microenvironment and challenges in delivering brain-penetrant agents, as highlighted by Touat et al. [28]. The persistent lack of effective systemic therapies highlights the substantial challenges that remain in achieving meaningful therapeutic breakthroughs for GBM.

While increased uptake of combined Sx+RT+Chemo reflects overall progress in the real-world treatment of GBM, persistent inequities in access remain due to demographic, socioeconomic, and systemic factors. In this study, multivariable analyses revealed that females had 10% lower odds of receiving combined Sx+RT+Chemo compared to males (OR: 0.90, 95% CI: 0.88–0.92, *p* < 0.001). Non-White patients also had lower odds of receiving combination therapy (OR: 0.90, 95% CI: 0.86–0.94, *p* < 0.001). Despite disparities in treatment access, Non-White and Hispanic patients demonstrated significantly better adjusted survival compared with White and non-Hispanic patients, respectively. These findings align with those observed by Reihanian et al., where women exhibited a higher median survival than men (16.14 months vs. 10.75 months, *p* = 0.023) [35]. Patients who did not receive Sx+RT+Chemo were likely to have been influenced by factors such as poor performance status or high comorbidity burden, limiting eligibility for aggressive multimodal therapy [36]. Geographic disparities were also evident, with patients in rural or underserved regions facing greater barriers to combination therapy receipt. Prior studies have noted that counties with higher Black populations and counties located in the Midwest (OR: 3.042) or West (OR: 3.175) were more likely to experience surgical delays [36]. These findings highlight opportunities for expanding telemedicine infrastructure, strengthening regional referral networks, and increasing investment in the neurosurgical workforce to enhance equitable access to standard-of-care therapies [37].

While these disparities are well-documented in the U.S. healthcare system, recent studies suggest that inequities in cancer treatment are also present in countries with universal healthcare models. For instance, Coppini et al. reported that patients in Central and Eastern Europe encounter barriers related to health misinformation and structural inefficiencies, despite broad coverage frameworks [38]. Similarly, Ferraris et al. highlighted persistent socioeconomic disparities in access and treatment navigation among Italian cancer patients [39]. Berchet et al. further emphasized that pan-European gaps in prevention and care contribute to unequal outcomes even within predominantly Caucasian populations [40]. Disparities in cancer treatment are not solely products of insurance status or racial demographics but can persist due to cultural, structural, and geographic challenges across healthcare systems.

Socioeconomic status was a significant determinant of treatment access. Patients residing in ZIP codes with median household incomes < $74,063 had lower odds of receiving combined Sx+RT+Chemo compared with patients from higher-income areas (≥$74,063) (OR: 1.15, 95% CI: 1.12–1.19, *p* < 0.001). This finding is supported by prior studies, where patients treated at safety-net hospitals were less likely to complete radiation or systemic therapy compared to those obtaining private care [41,42]. Insurance status further contributed to disparities: patients with government insurance (Medicare/Medicaid) and uninsured patients had significantly lower odds of receiving triple-modality therapy compared to privately insured patients [43,44]. Brown et al. similarly reported reduced odds of receiving surgery among Medicaid and uninsured patients (ORs: 0.72 and 0.77, respectively), with associated gaps in access to chemotherapy and radiation therapy [45].

Beyond standard oncologic management, anticoagulation in GBM remains an area of clinical complexity due to the competing risks of thromboembolic events and intracerebral hemorrhage. As highlighted by Bianconi et al., patients with high-grade gliomas face increased thrombotic risks, although prophylactic anticoagulation strategies must carefully weigh bleeding risks [46]. Future real-world studies should integrate data on anticoagulant management to further characterize its impact on GBM outcomes.

Survival analyses demonstrated a paradoxical pattern: despite facing barriers to treatment, Non-White and Hispanic patients exhibited significantly better adjusted survival compared to White and non-Hispanic patients, respectively. Survival outcomes varied by tumor histology. Patients with giant cell GBM exhibited improved survival relative to those with conventional GBM (aHR: 0.81, *p* < 0.001), reflecting the less infiltrative biology of this variant. Gliosarcoma, however, did not demonstrate a significant difference in survival compared to GBM NOS.

The impact of comorbidity burden on survival was notable. A stepwise increase in mortality risk was observed with higher Charlson–Deyo comorbidity scores. Patients with a CCI of 3 had a 44% higher risk of death compared to patients with no comorbidities (HR: 1.44, 95% CI: 1.40–1.48, *p* < 0.001). These results partly explain why survival gains observed in real-world populations were more modest compared to clinical trial populations. For instance, while Stupp et al. reported median OS of 14–16 months with standard chemoradiation, median OS in this cohort increased only from 8.2 to 9.8 months between 2005 and 2020. This discrepancy highlights the role of patient heterogeneity, access disparities, and real-world treatment limitations often underrepresented in trial populations.

### 4.1. Limitations

This study has several important limitations. First, the retrospective nature of the NCDB introduces inherent selection bias and restricts the ability to infer causality. The reliance on hospital-reported data from Commission on Cancer-accredited facilities limits generalizability, as non-hospital settings and international populations are not depicted [47]. Additionally, granular treatment-level details, such as specific chemotherapy agents (e.g., temozolomide), radiation dosages, or number of therapy cycles, are unavailable, precluding definitive assessment of adherence to full Stupp protocol standards.

Second, patient-reported outcomes, quality-of-life measures, and functional performance status are absent from the NCDB. As a result, the relationship between treatment receipt and broader patient well-being could not be evaluated. Third, while socioeconomic data are available at the ZIP code level, individual-level socioeconomic status and healthcare access measures (e.g., transportation barriers, specialist availability) are lacking. Differences in healthcare infrastructure, referral patterns, and hospital resources could not be fully delineated.

Additionally, during the 15-year study period, administrative coding transitioned from ICD-9 to ICD-10 classifications in 2016 for comorbidity and administrative variables, although tumor morphology coding (ICD-O-3) remained consistent. Due to the NCDB data structure, only aggregated Charlson–Deyo scores were available; specific comorbid conditions contributing to higher scores (e.g., CCI = 3) could not be individually assessed. While comorbidity burden was associated with worse survival outcomes, the impact of individual comorbid conditions could not be separately analyzed due to limitations in NCDB data granularity. Observed survival differences across facility types may partly reflect differences in access to specialized therapies, clinical trials, and supportive care resources, although these variables were not directly tabulated in the NCDB.

Finally, the increasing number of GBM cases reported over time is likely to reflect expansion of NCDB coverage through hospital accreditation rather than a true rise in disease incidence, a point that must be interpreted cautiously.

### 4.2. Implications and Future Research Directions

The findings of this study highlight persistent disparities in GBM treatment access and survival outcomes despite overall improvements in multimodal therapy delivery. Efforts to bridge these gaps must prioritize expanding telemedicine services, strengthening regional neuro-oncology networks, and enhancing access to specialized care, particularly in underserved rural and low-income areas.

Future research should focus on integrating molecular and genomic profiling into large datasets to enable precision medicine approaches for GBM. Brain-penetrant therapies, targeted agents, and novel delivery methods should be incorporated into combination regimens to address the challenges of the blood–brain barrier and tumor invasiveness. Prospective studies incorporating patient-reported outcomes and quality-of-life measures are essential to better understand the full impact of treatment beyond survival alone.

In addition, future analyses should explore disparities in clinical trial participation and therapeutic innovations across different healthcare systems. Comparative studies between U.S. and European populations, where universal healthcare systems may influence disparity patterns, could offer valuable insights into system-level drivers of inequity. Finally, designing inclusive, representative clinical trials with diverse socioeconomic and racial/ethnic enrollment will be critical to ensuring equitable therapeutic advancements for GBM patients worldwide.

## 5. Conclusions

This study highlights a significant increase in the adoption of combination therapy with Sx+RT+Chemo for GBM over a 15-year period, accompanied by modest improvements in survival. Despite these gains, median overall survival for combination therapy recipients remained limited at 14.6 months, reflecting the inherent aggressiveness of GBM. Persistent disparities were evident, with women, racial and ethnic minorities, and patients from lower-income regions less likely to receive triple-modality treatment. These inequities are likely to stem from differences in healthcare infrastructure, resource allocation, and systemic biases. The underutilization of molecular profiling and its capture in national datasets further constrains efforts to personalize therapy and optimize outcomes in real-world settings. Future efforts must prioritize the integration of emerging therapies with a commitment to equitable access. Reducing structural disparities and advancing precision medicine approaches will be essential to achieving meaningful, transformative improvements in GBM care.

## Figures and Tables

**Figure 1 brainsci-15-00556-f001:**
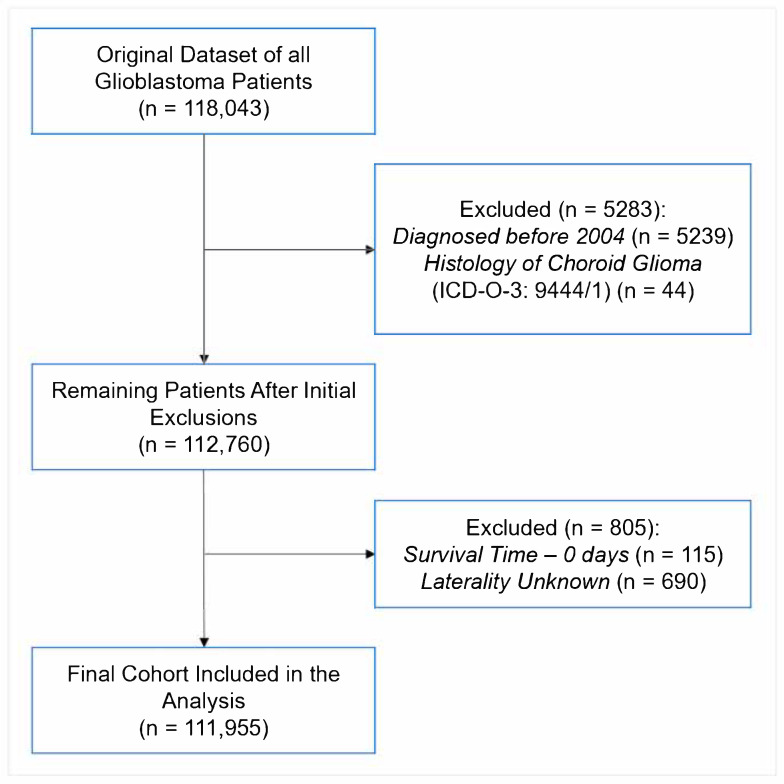
STROBE Diagram of Patient Inclusion for Analysis.

**Figure 2 brainsci-15-00556-f002:**
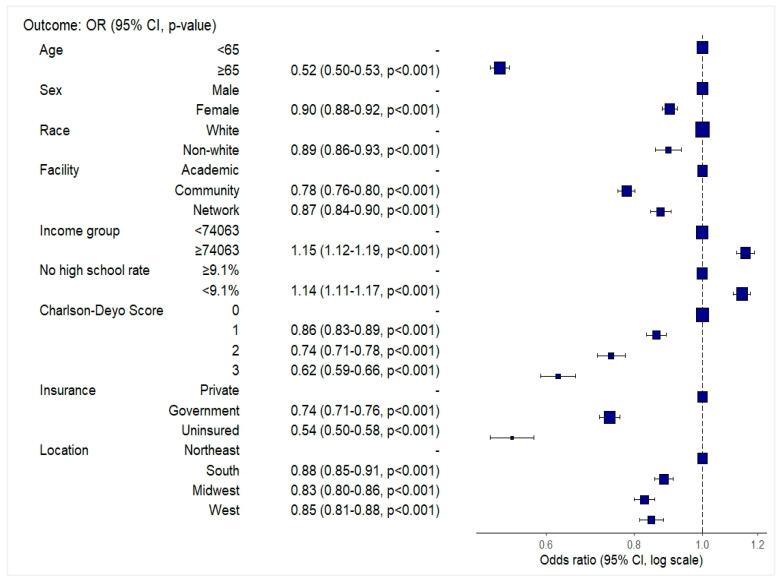
Forest plot depicting odds ratios and 95% confidence intervals for receipt of combined surgery, radiation, and chemotherapy among glioblastoma patients, based on multivariable logistic regression.

**Figure 3 brainsci-15-00556-f003:**
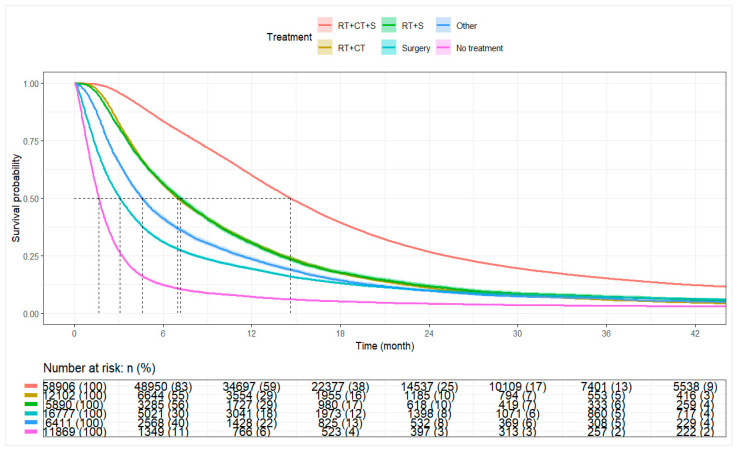
Kaplan–Meier curve for overall survival by treatment categories. **Abbreviations:** RT+CT+S: Radiation, chemotherapy, and surgery; RT+CT: Radiation and chemotherapy; RT+S: Radiation and surgery; Surgery: Surgery alone; No Treatment: No radiation, chemotherapy, or surgery.

**Figure 4 brainsci-15-00556-f004:**
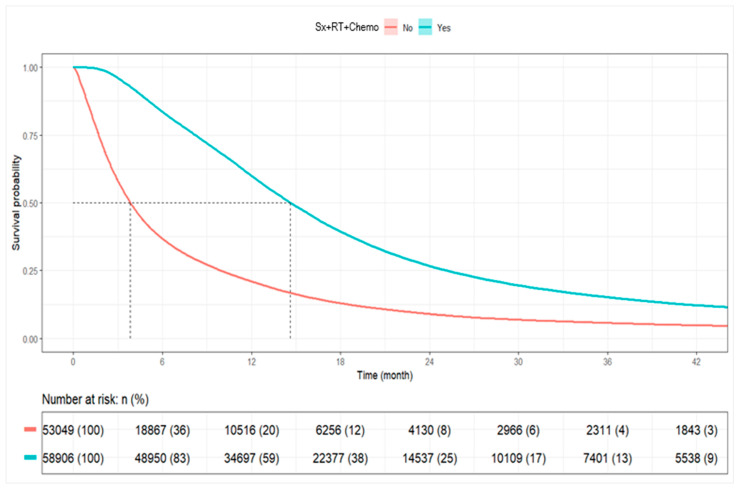
Kaplan–Meier Plot for Overall Survival by Combined RT+CT+S vs. Other. **Abbreviations:** Sx+RT+Chemo: Surgery, radiation, and chemotherapy.

**Table 1 brainsci-15-00556-t001:** Demographic and Disease Characteristics.

Year	Levels	All	2005–2008	2009–2012	2013–2016	2017–2020
**Age**	40–49	9918 (8.9)	2417 (10.9)	2488 (9.5)	2451 (8.1)	2562 (7.7)
	50–64	42,978 (38.4)	8616 (38.7)	10,431 (39.7)	11,667 (38.7)	12,264 (36.9)
	65–74	33,672 (30.1)	6061 (27.2)	7424 (28.3)	9159 (30.4)	11,028 (33.1)
	74–90	25,387 (22.7)	5182 (23.3)	5917 (22.5)	6867 (22.8)	7421 (22.3)
**Sex**	Male	64,392 (57.5)	12,654 (56.8)	14,965 (57.0)	17,390 (57.7)	19,383 (58.3)
	Female	47,563 (42.5)	9622 (43.2)	11,295 (43.0)	12,754 (42.3)	13,892 (41.7)
**Race**	White	102,170 (91.3)	20,681 (92.8)	24,176 (92.1)	27,470 (91.1)	29,843 (89.7)
	Non-white	9785 (8.7)	1595 (7.2)	2084 (7.9)	2674 (8.9)	3432 (10.3)
**Ethnicity**	Non-Hispanic	105,985 (94.7)	21,273 (95.5)	24,966 (95.1)	28,595 (94.9)	31,151 (93.6)
	Hispanic	5970 (5.3)	1003 (4.5)	1294 (4.9)	1549 (5.1)	2124 (6.4)
**Facility**	Academic	46,286 (41.3)	8542 (38.3)	10,537 (40.1)	12,701 (42.1)	14,506 (43.6)
	Community	43,630 (39.0)	9269 (41.6)	10,537 (40.1)	11,697 (38.8)	12,127 (36.4)
	Network	22,039 (19.7)	4465 (20.0)	5186 (19.7)	5746 (19.1)	6642 (20.0)
**Income Group ***	<$74,063	66,991 (59.8)	13,797 (61.9)	15,921 (60.6)	17,985 (59.7)	19,288 (58.0)
	≥$74,063	44,964 (40.2)	8479 (38.1)	10,339 (39.4)	12,159 (40.3)	13,987 (42.0)
**No High School Rate ****	≥9.1%	50,136 (44.8)	10,181 (45.7)	11,976 (45.6)	13,518 (44.8)	14,461 (43.5)
	<9.1%	61,819 (55.2)	12,095 (54.3)	14,284 (54.4)	16,626 (55.2)	18,814 (56.5)
**Region**	Metro	94,101 (84.1)	18,610 (83.5)	22,011 (83.8)	25,301 (83.9)	28,179 (84.7)
	Non-metro	17,854 (15.9)	3666 (16.5)	4249 (16.2)	4843 (16.1)	5096 (15.3)
**Charlson**–**Deyo Score *****	0	77,866 (69.6)	15,719 (70.6)	18,015 (68.6)	20,851 (69.2)	23,281 (70.0)
	1	19,629 (17.5)	3939 (17.7)	4880 (18.6)	5547 (18.4)	5263 (15.8)
	2	8926 (8.0)	1745 (7.8)	2187 (8.3)	2398 (8.0)	2596 (7.8)
	3	5534 (4.9)	873 (3.9)	1178 (4.5)	1348 (4.5)	2135 (6.4)
**Insurance**	Private	45,912 (41.0)	10,036 (45.1)	11,258 (42.9)	12,184 (40.4)	12,434 (37.4)
	Government	62,614 (55.9)	11,498 (51.6)	14,001 (53.3)	17,122 (56.8)	19,993 (60.1)
	Uninsured	3429 (3.1)	742 (3.3)	1001 (3.8)	838 (2.8)	848 (2.5)
**Location**	Northeast	41,185 (36.8)	8292 (37.2)	9643 (36.7)	11,257 (37.3)	11,993 (36.0)
	South	31,904 (28.5)	6587 (29.6)	7598 (28.9)	8432 (28.0)	9287 (27.9)
	Midwest	22,717 (20.3)	4210 (18.9)	5202 (19.8)	6143 (20.4)	7162 (21.5)
	West	16,149 (14.4)	3187 (14.3)	3817 (14.5)	4312 (14.3)	4833 (14.5)
**Histology**	Glioblastoma, NOS	108,426 (96.8)	21,664 (97.3)	25,469 (97.0)	29,255 (97.1)	32,038 (96.3)
	Giant cell glioblastoma	766 (0.7)	154 (0.7)	199 (0.8)	238 (0.8)	175 (0.5)
	Gliosarcoma	2377 (2.1)	458 (2.1)	592 (2.3)	651 (2.2)	676 (2.0)
	Glioblastoma, IDH-mutant	386 (0.3)	0 (0.0)	0 (0.0)	0 (0.0)	386 (1.2)
**Laterality ******	Unilateral	82,791 (74.0)	16,451 (73.9)	19,293 (73.5)	22,340 (74.1)	24,707 (74.3)
	Bilateral/Midline	29,164 (26.0)	5825 (26.1)	6967 (26.5)	7804 (25.9)	8568 (25.7)
**Primary Site**	Frontal-temporal	58,338 (52.1)	11,182 (50.2)	13,412 (51.1)	15,912 (52.8)	17,832 (53.6)
	Parietal-occipital	21,993 (19.6)	4770 (21.4)	5303 (20.2)	5737 (19.0)	6183 (18.6)
	Overlapping lesion	15,948 (14.2)	3522 (15.8)	3851 (14.7)	4118 (13.7)	4457 (13.4)
	Ventricle-cerebellum	994 (0.9)	193 (0.9)	219 (0.8)	253 (0.8)	329 (1.0)
	Brain stem	363 (0.3)	72 (0.3)	82 (0.3)	120 (0.4)	89 (0.3)
	Brain NOS	10,164 (9.1)	1714 (7.7)	2460 (9.4)	2838 (9.4)	3152 (9.5)
	Cerebrum	4155 (3.7)	823 (3.7)	933 (3.6)	1166 (3.9)	1233 (3.7)

**Abbreviations:** GBM: Glioblastoma Multiforme; IDH: Isocitrate Dehydrogenase; NOS: Not Otherwise Specified; HS: High School; Charlson–Deyo Score: Comorbidity index based on ICD codes; n (%): Frequency and percentage; Metro: Metropolitan area; Non-metro: Non-metropolitan area. * Income group was determined based on the median household income in the patient’s ZIP code of residence, categorized as <$74,063 vs. ≥$74,063. ** High school education level reflects the percentage of adults without a high school diploma in the ZIP code of residence. Categories: ≥9.1% (high non-completion), <9.1% (low non-completion). *** Charlson–Deyo Score: 0 = no comorbid conditions; 1 = one comorbid condition; 2 = two comorbid conditions; 3 = three or more comorbid conditions. **** Laterality: Unilateral = tumor confined to one hemisphere; Bilateral/Midline = tumor crossing or located at midline structures.

**Table 2 brainsci-15-00556-t002:** Treatment Patterns and Outcomes Across Four Time Periods.

Variable	Evaluable Population (N)	Category/Level	2005–2008 N (%)	2009–2012 N (%)	2013–2016 N (%)	2017–2020 N (%)
**Chemotherapy**	111,955	No	8286 (37.2)	8717 (33.2)	9912 (32.9)	10,557 (31.7)
		Yes	13,990 (62.8)	17,543 (66.8)	20,232 (67.1)	22,718 (68.3)
**Radiation Therapy**	111,955	No	6739 (30.3)	7548 (28.7)	8567 (28.4)	9267 (27.8)
		Yes	15,537 (69.7)	18,712 (71.3)	21,577 (71.6)	24,008 (72.2)
**Surgical Treatment**	111,955	No	6176 (27.7)	6625 (25.2)	7251 (24.1)	7584 (22.8)
		Yes	16,100 (72.3)	19,635 (74.8)	22,893 (75.9)	25,691 (77.2)
**Immunotherapy**	111,773	No	22,135 (99.7)	26,065 (99.5)	28,922 (96.0)	31,152 (93.7)
		Yes	65 (0.3)	135 (0.5)	1200 (4.0)	2099 (6.3)
**Days from Diagnosis to Radiation Start**	78,284	Median (IQR)	29.0 (21.0 to 39.0)	32.0 (24.0 to 42.0)	34.0 (27.0 to 44.0)	36.0 (28.0 to 47.0)
**30-day Mortality**	83,668	Died <30 days	998 (6.2)	967 (5.0)	1100 (4.8)	1095 (4.3)
		Alive ≥30 days	15,001 (93.8)	18,560 (95.0)	21,642 (95.2)	24,305 (95.7)
**90-day Mortality**	83,444	Died <90 days	2877 (18)	3053 (15.6)	3425 (15.1)	3658 (14.5)
		Alive ≥90 days	13,107 (82)	16,456 (84.4)	19,263 (84.9)	21,605 (85.5)
**30-Day Re-Admission**	111,955	No	21,027 (94.4)	24,776 (94.3)	28,485 (94.5)	31,534 (94.8)
		Yes	1249 (5.6)	1484 (5.7)	1659 (5.5)	1741 (5.2)
**Surgical Discharge Days**	73,943	Median (IQR)	4.0 (3.0 to 7.0)	4.0 (2.0 to 6.0)	3.0 (2.0 to 6.0)	3.0 (2.0 to 6.0)
**Elapsed Days for Radiation Therapy**	76,235	Median (IQR)	42.0 (0.0 to 46.0)	42.0 (0.0 to 44.0)	41.0 (0.0 to 44.0)	37.0 (0.0 to 43.0)
**Overall Survival (months)**	111,955	Median (IQR)	7.8 (3.0 to 16.7)	8.9 (3.4 to 18.7)	9.2 (3.4 to 18.8)	9.5 (3.5 to 18.6)

**Table 3 brainsci-15-00556-t003:** Median OS and Survival Rates by Treatment Category.

Treatment	N	Median (Months)	3-Month OS Rate (%)	6-Month OS Rate (%)	1-Year OS Rate (%)	2-Year OS Rate (%)	3-Year OS Rate (%)
**All**	111,955	9.30 (9.20–9.40)	78.2 (77.9–78.4)	61.6 (61.3–61.9)	41.6 (41.4–41.9)	18.4 (18.2–18.6)	10.8 (10.6–11.0)
**RT+CT+S**	58,906	14.62 (14.52–14.75)	96.0 (95.9–96.2)	83.6 (83.3–83.9)	59.9 (59.5–60.3)	26.7 (26.3–27.0)	15.2 (14.9–15.5)
**RT+CT**	12,102	7.00 (6.80–7.20)	82.6 (82.0–83.3)	56.0 (55.1–56.9)	30.6 (29.8–31.5)	10.9 (10.4–11.5)	5.8 (5.4–6.3)
**RT+S**	5890	7.16 (6.90–7.39)	80.9 (79.9–81.9)	56.5 (55.2–57.7)	30.3 (29.1–31.5)	11.8 (11.0–12.7)	7.1 (6.5–7.9)
**Surgery**	16,777	3.09 (3.00–3.15)	50.7 (50.0–51.5)	31.0 (30.3–31.7)	19.2 (18.6–19.8)	9.8 (9.4–10.3)	7.0 (6.6–7.4)
**Other**	6411	4.57 (4.44–4.73)	65.6 (64.4–66.8)	41.2 (40.0–42.5)	23.5 (22.4–24.5)	9.8 (9.0–10.6)	6.3 (5.7–7.0)
**No treatment**	11,869	1.64 (1.61–1.68)	27.0 (26.2–27.8)	12.2 (11.6–12.9)	7.1 (6.6–7.6)	4.1 (3.7–4.5)	3.2 (2.9–3.6)

**Abbreviations:** Sx = Surgery; RT = Radiation Therapy; Chemo = Chemotherapy; OS = Overall Survival.

**Table 4 brainsci-15-00556-t004:** Median Overall Survival by Combined RT+CT+S vs. Other.

Combination Sx+RT+Chemo	N	Median (Months)	3-Month OS Rate (%)	6-Month OS Rate (%)	1-Year OS Rate (%)	2-Year OS Rate (%)	3-Year OS Rate (%)
**All**	111,955	9.30 (9.20–9.40)	78.2 (77.9–78.4)	61.6 (61.3–61.9)	41.6 (41.4–41.9)	18.4 (18.2–18.6)	10.8 (10.6–11.0)
**No**	53,049	3.84 (3.81–3.91)	58.1 (57.7–58.5)	36.8 (36.4–37.2)	21.0 (20.6–21.3)	9.0 (8.8–9.3)	5.8 (5.6–6.0)
**Yes**	58,906	14.62 (14.52–14.75)	96.0 (95.9–96.2)	83.6 (83.3–83.9)	59.9 (59.5–60.3)	26.7 (26.3–27.0)	15.2 (14.9–15.5)

**Abbreviations:** Sx = Surgery; RT = Radiation Therapy; Chemo = Chemotherapy; OS = Overall Survival.

**Table 5 brainsci-15-00556-t005:** Cox Proportional Hazard Models for Overall Survival.

Variable	Levels	n (%)	HR (Univariable)	HR (Multivariable)
**Treatment ^**	Sx+RT+Chemo	58,906 (52.6)	-	-
	RT+Chemo	12,102 (10.8)	1.74 (1.70–1.77, *p* < 0.001)	1.66 (1.62–1.69, *p* < 0.001)
	RT+Sx	5890 (5.3)	1.69 (1.64–1.74, *p* < 0.001)	1.60 (1.55–1.64, *p* < 0.001)
	Sx	16,777 (15.0)	2.47 (2.43–2.52, *p* < 0.001)	2.36 (2.31–2.40, *p* < 0.001)
	Other	6411 (5.7)	2.12 (2.06–2.18, *p* < 0.001)	2.02 (1.96–2.07, *p* < 0.001)
	No treatment	11,869 (10.6)	5.07 (4.96–5.18, *p* < 0.001)	4.38 (4.29–4.48, *p* < 0.001)
**Age**	<65	52,896 (47.2)	-	-
	≥65	59,059 (52.8)	1.82 (1.80–1.85, *p* < 0.001)	1.43 (1.41–1.46, *p* < 0.001)
**Sex**	Male	64,392 (57.5)	-	-
	Female	47,563 (42.5)	0.99 (0.98–1.00, *p* = 0.058)	0.95 (0.94–0.96, *p* < 0.001)
**Race**	White	102,170 (91.3)	-	-
	Non-white	9785 (8.7)	0.83 (0.81–0.85, *p* < 0.001)	0.78 (0.76–0.80, *p* < 0.001)
**Ethnicity**	Non-Hispanic	105,985 (94.7)	-	-
	Hispanic	5970 (5.3)	0.80 (0.77–0.82, *p* < 0.001)	0.75 (0.73–0.77, *p* < 0.001)
**Facility**	Academic	46,286 (41.3)	-	-
	Community	43,630 (39.0)	1.24 (1.23–1.26, *p* < 0.001)	1.09 (1.07–1.10, *p* < 0.001)
	Network	22,039 (19.7)	1.18 (1.16–1.20, *p* < 0.001)	1.12 (1.11–1.14, *p* < 0.001)
**Income group ***	<$74,063	66,991 (59.8)	-	-
	≥$74,063	44,964 (40.2)	0.83 (0.82–0.84, *p* < 0.001)	0.86 (0.85–0.87, *p* < 0.001)
**No high school rate ****	≥9.1%	50,136 (44.8)	-	-
	<9.1%	61,819 (55.2)	0.94 (0.92–0.95, *p* < 0.001)	1.05 (1.04–1.07, *p* < 0.001)
**Region**	Metro	94,101 (84.1)	-	-
	Non-metro	17,854 (15.9)	1.16 (1.14–1.18, *p* < 0.001)	1.06 (1.04–1.08, *p* < 0.001)
**Charlson–Deyo Score *****	0	77,866 (69.6)	-	-
	1	19,629 (17.5)	1.28 (1.26–1.31, *p* < 0.001)	1.18 (1.16–1.20, *p* < 0.001)
	2	8926 (8.0)	1.41 (1.38–1.45, *p* < 0.001)	1.28 (1.25–1.31, *p* < 0.001)
	3	5534 (4.9)	1.67 (1.62–1.72, *p* < 0.001)	1.44 (1.40–1.48, *p* < 0.001)
**Insurance**	Private	45,912 (41.0)	-	-
	Government	62,614 (55.9)	1.67 (1.65–1.69, *p* < 0.001)	1.15 (1.13–1.17, *p* < 0.001)
	Uninsured	3429 (3.1)	1.14 (1.09–1.18, *p* < 0.001)	1.01 (0.97–1.05, *p* = 0.610)
**Location**	Northeast	41,185 (36.8)	-	-
	South	31,904 (28.5)	1.11 (1.09–1.13, *p* < 0.001)	1.04 (1.03–1.06, *p* < 0.001)
	Midwest	22,717 (20.3)	1.04 (1.02–1.05, *p* < 0.001)	0.95 (0.93–0.96, *p* < 0.001)
	West	16,149 (14.4)	1.04 (1.02–1.06, *p* < 0.001)	1.06 (1.04–1.08, *p* < 0.001)
**Histology**	Glioblastoma, NOS	108,426 (96.8)	-	-
	Giant cell glioblastoma	766 (0.7)	0.72 (0.67–0.78, *p* < 0.001)	0.81 (0.75–0.88, *p* < 0.001)
	Gliosarcoma	2377 (2.1)	0.92 (0.88–0.96, *p* < 0.001)	1.03 (0.99–1.07, *p* = 0.179)
	Glioblastoma, IDH-mutant	386 (0.3)	0.45 (0.40–0.52, *p* < 0.001)	0.56 (0.50–0.64, *p* < 0.001)
**Laterality ******	Unilateral	82,791 (74.0)	-	-
	Bilateral/midline	29,164 (26.0)	1.24 (1.23–1.26, *p* < 0.001)	1.12 (1.11–1.14, *p* < 0.001)

**Abbreviations:** Sx: Surgery; RT: Radiation Therapy; Chemo: Chemotherapy; HR: Hazard Ratio; NOS: Not Otherwise Specified; IDH: Isocitrate Dehydrogenase; Metro: Metropolitan area; Non-metro: Non-metropolitan area; Charlson–Deyo Score: Comorbidity index based on ICD codes; n (%): Frequency and percentage. * Income group was determined based on the median household income in the patient’s ZIP code of residence, categorized as <$74,063 vs. ≥$74,063. ** High school education level reflects the percentage of adults without a high school diploma in the ZIP code of residence. Categories: ≥9.1% (high non-completion), <9.1% (low non-completion). *** Charlson–Deyo Score: 0 = no comorbid conditions; 1 = one comorbid condition; 2 = two comorbid conditions; 3 = three or more comorbid conditions. ^ Treatment groups: (i) Sx+RT+Chemo = Radiation therapy + Chemotherapy + Surgery (reference group), (ii) RT+Chemo = Radiation therapy + Chemotherapy, (iii) RT+Sx = Radiation therapy + Surgery. **** Laterality: Unilateral = tumor confined to one hemisphere; Bilateral/Midline = tumor crossing or located at midline structures.

**Table 6 brainsci-15-00556-t006:** Multivariable Weibull Accelerated Failure Time Model for Overall Survival.

Variable	Category	Time Ratio (TR)	95% CI	*p*-Value
**Treatment**	Sx+RT+Chemo (Reference)	1.00	—	—
	RT+Chemo	0.58	0.57–0.59	<0.001
	RT+Surgery	0.62	0.61–0.63	<0.001
	Surgery alone	0.43	0.42–0.44	<0.001
	Other therapies	0.49	0.48–0.51	<0.001
	No treatment	0.24	0.23–0.24	<0.001
**Age**	<65 years (Reference)	1.00	—	—
	≥65 years	0.67	0.66–0.68	<0.001
**Sex**	Male (Reference)	1.00	—	—
	Female	1.07	1.06–1.08	<0.001
**Race**	White (Reference)	1.00	—	—
	Non-White	1.33	1.30–1.36	<0.001
**Ethnicity**	Non-Hispanic (Reference)	1.00	—	—
	Hispanic	1.40	1.34–1.46	<0.001
**Facility Type**	Academic (Reference)	1.00	—	—
	Community	0.91	0.90–0.92	<0.001
	Network	0.87	0.86–0.88	<0.001
**Income Group**	<$74,063 (Reference)	1.00	—	—
	≥$74,063	1.19	1.18–1.20	<0.001
**Education**	≥9.1% no HS diploma (Reference)	1.00	—	—
	<9.1% no HS diploma	0.94	0.93–0.95	<0.001
**Region Type**	Metropolitan (Reference)	1.00	—	—
	Non-metropolitan	0.93	0.91–0.95	<0.001
**Charlson–Deyo Score**	0 (Reference)	1.00	—	—
	1	0.84	0.83–0.85	<0.001
	2	0.77	0.75–0.79	<0.001
	3	0.67	0.65–0.69	<0.001
**Insurance**	Private (Reference)	1.00	—	—
	Government	0.85	0.84–0.86	<0.001
	Uninsured	1.01	0.96–1.06	0.610
**Region**	Northeast (Reference)	1.00	—	—
	South	0.94	0.92–0.95	<0.001
	Midwest	1.05	1.03–1.07	<0.001
	West	0.91	0.90–0.93	<0.001
**Histology**	Glioblastoma, NOS (Reference)	1.00	—	—
	Giant cell glioblastoma	1.36	1.26–1.47	<0.001
	Gliosarcoma	0.96	0.90–1.02	0.118
	Glioblastoma, IDH-mutant	1.70	1.48–1.95	<0.001
**Laterality**	Unilateral (Reference)	1.00	—	—
	Bilateral/Midline	0.89	0.88–0.90	<0.001

**Abbreviations:** Sx: Surgery; RT: Radiation Therapy; Chemo: Chemotherapy; TR: Time Ratio; CI: Confidence Interval; NOS: Not Otherwise Specified; IDH: Isocitrate Dehydrogenase; HS: High School; Charlson–Deyo Score: Comorbidity index based on ICD codes.

## Data Availability

The data supporting the findings of this study are available from the National Cancer Database.

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
