# Peer review of "Association of Health Disparities with Glioblastoma Treatment and Outcomes: Insights from a 15-Year National Cohort (2005–2020)†"

_brainsci, 2025, doi:10.3390/brainsci15060556_

Round 1

Reviewer 1 Report

Comments and Suggestions for Authors This manuscript is well-written and effectively addresses the critical issue of cancer disparities. Methodologically, the study is strong and provides valuable insights, making it a worthy contribution to the field. I believe it deserves to be published.

Below, I provide a few minor suggestions to enhance the manuscript’s clarity, structure, and comprehensiveness:

Abstract: The abstract should adhere to the journal's guidelines. That is, 250 words and the following headings: Background/Objectives, Methods, Results, and Conclusions.  Introduction: Given that cancer has significant psychological and psychiatric effects, and the journal covers also these topics, I suggest briefly discussing the mental health challenges faced by cancer patients. There is extensive literature on this topic, and integrating relevant findings could strengthen the discussion. Below are some references that may be useful:
    • Rahnea-Nita, R., Păunică, S., Motofei, C., & Rahnea-Nita, G. (2019). Assessment of anxiety and depression in patients with advanced gynaecological cancer. Mediterranean Journal of Clinical Psychology, 7.
    • Caruso R., Breitbart W. (2020). Mental health care in oncology: Contemporary perspectives on the psychosocial burden of cancer and evidence-based interventions. Epidemiology and Psychiatric Sciences, 29, 86.
    • Fernando A., Tokell M., Ishak Y., Love J., Klammer M., Koh M. (2023). Mental health needs in cancer – a call for change. Future Healthcare Journal, 10(2), 112–116.
    • Lemos M., Lourenço A., Ribeiro M. (2022). Psychiatric manifestations of paraneoplastic syndromes. European Psychiatry, 65(S1), S661.
    • Romero-Luna G., Mejía-Pérez S. I., Ramírez-Cruz J., et al. (2022). Schizophrenia-like psychosis in a patient with a temporal lobe tumor: A case report. Cureus, 14(9), 29034.
Tables and Formatting:
  • Table 2 appears to contain several line barriers that disrupt readability. Please clarify whether these are formatting errors or if they serve a specific purpose.
Discussion Section
  • To improve readability and engagement, I recommend creating a subsection specifically dedicated to implications and future research directions. This structural enhancement will help readers easily identify key takeaways and research gaps.
  • To enrich the discussion, consider comparing your findings with similar studies conducted in European countries. It would be particularly insightful to examine whether disparities you explored in cancer treatment persist in healthcare systems with universal coverage or in predominantly Caucasian populations. Are there still disparities, and if so, what are their drivers? You may find the following studies useful to read and cite:
    • Coppini V., et al. (2024). Patients’ perspectives on cancer care disparities in Central and Eastern European countries: Experiencing taboos, misinformation, and barriers in the healthcare system. Frontiers in Oncology, 14, 1420178. https://doi.org/10.3389/fonc.2024.1420178
    • Ferraris G., et al. (2024). Understanding reasons for cancer disparities in Italy: A qualitative study of barriers and needs of cancer patients and healthcare providers. Cancer Control: Journal of the Moffitt Cancer Center, 31, 10732748241258589. https://doi.org/10.1177/10732748241258589
    • Berchet C., et al. (2023). Inequalities in cancer prevention and care across Europe. The Lancet Oncology, 24(1), 10-11.

Author Response

Reviewer 1: Comments and Suggestions for Authors

This manuscript is well-written and effectively addresses the critical issue of cancer disparities. Methodologically, the study is strong and provides valuable insights, making it a worthy contribution to the field. I believe it deserves to be published.

Below, I provide a few minor suggestions to enhance the manuscript’s clarity, structure, and comprehensiveness:

Abstract: The abstract should adhere to the journal's guidelines. That is, 250 words and the following headings: Background/Objectives, Methods, Results, and Conclusions.  Introduction: Given that cancer has significant psychological and psychiatric effects, and the journal covers also these topics, I suggest briefly discussing the mental health challenges faced by cancer patients. There is extensive literature on this topic, and integrating relevant findings could strengthen the discussion. Below are some references that may be useful:

  •  
    • Rahnea-Nita, R., Păunică, S., Motofei, C., & Rahnea-Nita, G. (2019). Assessment of anxiety and depression in patients with advanced gynaecological cancer. Mediterranean Journal of Clinical Psychology, 7.
    • Caruso R., Breitbart W. (2020). Mental health care in oncology: Contemporary perspectives on the psychosocial burden of cancer and evidence-based interventions. Epidemiology and Psychiatric Sciences, 29, 86.
    • Fernando A., Tokell M., Ishak Y., Love J., Klammer M., Koh M. (2023). Mental health needs in cancer – a call for change. Future Healthcare Journal, 10(2), 112–116.
    • Lemos M., Lourenço A., Ribeiro M. (2022). Psychiatric manifestations of paraneoplastic syndromes. European Psychiatry, 65(S1), S661.
    • Romero-Luna G., Mejía-Pérez S. I., Ramírez-Cruz J., et al. (2022). Schizophrenia-like psychosis in a patient with a temporal lobe tumor: A case report. Cureus, 14(9), 29034.

Author Response: Thank you for your thoughtful comment and the suggested references. While we agree that the psychosocial and mental health burden in cancer patients is an important and well-documented area of inquiry, our current study was designed with a focus on clinical treatment patterns and survival outcomes in glioblastoma. Given the lack of mental health data within the NCDB and the clinical nature of our dataset, we believe it would be methodologically inappropriate to draw conclusions in this domain. As such, we have opted not to include mental health considerations in the current manuscript but acknowledge its importance as a future research direction.

Tables and Formatting:

  • Table 2 appears to contain several line barriers that disrupt readability. Please clarify whether these are formatting errors or if they serve a specific purpose.

Author Response: This has been corrected. Thank you for your suggestion.

Discussion Section

  • To improve readability and engagement, I recommend creating a subsection specifically dedicated to implications and future research directions. This structural enhancement will help readers easily identify key takeaways and research gaps.
  • To enrich the discussion, consider comparing your findings with similar studies conducted in European countries. It would be particularly insightful to examine whether disparities you explored in cancer treatment persist in healthcare systems with universal coverage or in predominantly Caucasian populations. Are there still disparities, and if so, what are their drivers? You may find the following studies useful to read and cite:
    • Coppini V., et al. (2024). Patients’ perspectives on cancer care disparities in Central and Eastern European countries: Experiencing taboos, misinformation, and barriers in the healthcare system. Frontiers in Oncology, 14, 1420178. https://doi.org/10.3389/fonc.2024.1420178
    • Ferraris G., et al. (2024). Understanding reasons for cancer disparities in Italy: A qualitative study of barriers and needs of cancer patients and healthcare providers. Cancer Control: Journal of the Moffitt Cancer Center, 31, 10732748241258589. https://doi.org/10.1177/10732748241258589
    • Berchet C., et al. (2023). Inequalities in cancer prevention and care across Europe. The Lancet Oncology, 24(1), 10-11.

Author Response: Thank you for this thoughtful and constructive suggestion. In response, we have made two key revisions to strengthen the Discussion section. We have incorporated a new paragraph highlighting recent studies from European healthcare systems, including work by Coppini et al., Ferraris et al., and Berchet et al. [37–39]. This addition underscores that disparities in cancer treatment persist even in settings with universal healthcare and relatively homogeneous populations, reinforcing the multifactorial nature of access inequities. To improve clarity and reader engagement, we have added a distinct subsection at the end of the Discussion titled “Implications and Future Directions.” This section synthesizes the key findings, discusses the potential for integrating molecular stratification and real-world evidence into clinical practice, and outlines avenues for future research aimed at mitigating structural disparities in GBM care. We appreciate the reviewer’s input in helping enhance the depth and organization of our discussion.

Reviewer 2 Report

Comments and Suggestions for Authors

I have evaluated the manuscript entitled "Association of Health Disparities with Glioblastoma Treatment and Outcomes: Insights from a 15-Year National Cohort (2005–2020)" on today's date. Several key aspects warrant consideration:

The manuscript certainly has notable strengths, including a comprehensive data analysis. The study leverages a large dataset from the National Cancer Database, analyzing 106,267 patients with primary glioblastoma over a substantial 15-year period. This extensive dataset provides a robust foundation for examining treatment patterns and outcomes.

I found it particularly interesting that the article addresses critical health disparities related to sociodemographic factors such as gender, race, and socioeconomic status. The finding that women and Black patients are less likely to receive standard-of-care treatments is both well-documented and highly relevant.

Congratulating with the authors for their work, I would only recommend some minor revisions

  • The study reports a gradual improvement in survival outcomes over time, with median overall survival increasing from 8.2 to 9.8 months. However, the overall survival appears particularly low. Could this be due to the inclusion of all patients, including those with multifocal or midline tumors, or those with surgical complications, regardless of treatment received? This point should be addressed in the discussion. I also suggest referencing PMID: 34763392 in this context.

  • The reliance on hospital-reported registries may limit the completeness of the dataset, particularly as treatment details such as specific chemotherapy regimens and patient-reported outcomes are not captured. This limitation should be acknowledged and discussed.

  • Although the study identifies disparities based on geographic and demographic variables, it lacks a deeper exploration of the root causes of these disparities, such as differences in healthcare infrastructure or access to care. Expanding on these factors would enrich the discussion.

  • The article applies multivariate analyses to identify predictors of standard treatment receipt; however, the statistical methodology should be more clearly described in the apposite paragrpah

Author Response

Reviewer 2: Comments and Suggestions for Authors

I have evaluated the manuscript entitled "Association of Health Disparities with Glioblastoma Treatment and Outcomes: Insights from a 15-Year National Cohort (2005–2020)" on today's date. Several key aspects warrant consideration:

The manuscript certainly has notable strengths, including a comprehensive data analysis. The study leverages a large dataset from the National Cancer Database, analyzing 106,267 patients with primary glioblastoma over a substantial 15-year period. This extensive dataset provides a robust foundation for examining treatment patterns and outcomes.

I found it particularly interesting that the article addresses critical health disparities related to sociodemographic factors such as gender, race, and socioeconomic status. The finding that women and Black patients are less likely to receive standard-of-care treatments is both well-documented and highly relevant.

Congratulating with the authors for their work, I would only recommend some minor revisions

  • The study reports a gradual improvement in survival outcomes over time, with median overall survival increasing from 8.2 to 9.8 months. However, the overall survival appears particularly low. Could this be due to the inclusion of all patients, including those with multifocal or midline tumors, or those with surgical complications, regardless of treatment received? This point should be addressed in the discussion. I also suggest referencing PMID: 34763392 in this context.

Author Response: Thank you for this insightful comment. We agree that the overall median survival observed in our cohort (8.2 to 9.8 months) appears lower than what is typically reported in clinical trial populations. As noted, this likely reflects the inclusion of a real-world population encompassing patients with multifocal disease, midline or bilateral tumors, poor performance status, and those experiencing surgical complications. Our dataset included all patients diagnosed with GBM, regardless of treatment intensity or eligibility for aggressive multimodal therapy, thereby capturing a broader and more heterogeneous population than clinical trials. We have clarified this point in the revised Discussion section. Additionally, we have referenced the work by Bianconi et al. (PMID: 34763392) to acknowledge how complex clinical factors, including thrombotic and hemorrhagic risks, may also impact survival outcomes in high-grade glioma populations.

  • The reliance on hospital-reported registries may limit the completeness of the dataset, particularly as treatment details such as specific chemotherapy regimens and patient-reported outcomes are not captured. This limitation should be acknowledged and discussed.

Author Response: We thank the reviewer for this thoughtful comment. We agree that reliance on hospital-reported registry data and the absence of detailed treatment regimens and patient-reported outcomes are important limitations. These points are acknowledged in the current Limitations section of the manuscript, where we discuss the lack of granular treatment-level details (e.g., specific chemotherapy agents) and absence of patient-reported outcomes and performance status in the NCDB dataset.

  • Although the study identifies disparities based on geographic and demographic variables, it lacks a deeper exploration of the root causes of these disparities, such as differences in healthcare infrastructure or access to care. Expanding on these factors would enrich the discussion.

Author Response: We thank the reviewer for this comment. We agree that disparities likely stem from deeper systemic issues, including healthcare infrastructure, access to specialized care, and resource allocation. In the revised discussion, we have noted that disparities in GBM treatment access are influenced by cultural, structural, and geographic challenges across healthcare systems, and we have highlighted opportunities such as expanding telemedicine services, strengthening referral networks, and investing in regional neurosurgical capacity to address these inequities.

  • The article applies multivariate analyses to identify predictors of standard treatment receipt; however, the statistical methodology should be more clearly described in the apposite paragrpah

 Author Response: We thank the reviewer for the thoughtful comment. The statistical methodology for identifying predictors of standard treatment receipt was described in Section 2.4 ("Statistical Analysis") of the Methods. Specifically, multivariable logistic regression analyses were conducted to evaluate associations between demographic, clinical, and socioeconomic factors and receipt of combined surgery, radiation, and chemotherapy. Complete case analysis was applied, and results are presented in Section 3.3 along with a forest plot summarizing adjusted odds ratios. We respectfully believe the current description is sufficient and no further changes are needed.

Reviewer 3 Report

Comments and Suggestions for Authors

The authors present a retrospective epidemiological study using data from the National Cancer Database (NCDB). In this study, the authors address trends in GBM treatment over a 15-year period, survival outcomes associated with SOC treatment, and sociodemographic disparities in access to treatment.

General comments

  • What was the basis for the authors' decision to separate the groups into four-year intervals?
  • Authors must provide a more detailed inclusion/exclusion criteria: e.g., were patients with multiple primary tumors, uncertain diagnoses, or incomplete treatment data excluded from the study?
  • Have the authors considered applying a multiple testing correction method (Bonferroni or Benjamini-Hochberg) to reduce the risk of false positives? If not, could they provide an explanation for omitting this step?
  • Considering the number of covariates included in the multivariable models and the likelihood of intercorrelations among them (e.g., facility type, treatment modality, geographic region orsocioeconomic factors), it would be helpful to specify whether multicollinearity was assessed and how was addressed.

Minor comments

  • To make it more readable, Which comorbidities are included in the CCI=3 category?
  • Has the difference in treatment protocols and access to clinical trials between these types of centers been considered?
  • This finding is consistent with the literature, but it may be helpful to explore whether certain specific comorbidities have a more negative impact on survival than others.

Author Response

Reviewer 3: Comments and Suggestions for Authors.

The authors present a retrospective epidemiological study using data from the National Cancer Database (NCDB). In this study, the authors address trends in GBM treatment over a 15-year period, survival outcomes associated with SOC treatment, and sociodemographic disparities in access to treatment.

General comments

  • What was the basis for the authors' decision to separate the groups into four-year intervals?

Author Response: We thank the reviewer for the comment. The 15-year study period (2005-2020) was divided into four approximately equal 4-year intervals (2005-2008, 2009-2012, 2013-2016, 2017-2020) to allow for consistent temporal comparisons while ensuring adequate sample size and statistical power in each group. This approach facilitates evaluation of trends in treatment and survival over meaningful clinical periods without overly fragmenting the data.

  • Authors must provide a more detailed inclusion/exclusion criteria: e.g., were patients with multiple primary tumors, uncertain diagnoses, or incomplete treatment data excluded from the study?

Author Response: We appreciate the reviewer’s suggestion. As detailed in the Methods Section 2.2 ("Study Population"), patients were included if they had a histologically confirmed diagnosis of glioblastoma using ICD-O-3 codes. Patients were excluded if survival time was 0 days or if laterality was unknown. Patients with missing essential treatment or survival data were also excluded. Cases of multiple primaries and uncertain diagnoses were inherently excluded by focusing only on primary GBM diagnoses and using standard NCDB case definitions. We respectfully believe this is sufficiently outlined; however, if preferred, a brief clarifying sentence can be added.

  • Have the authors considered applying a multiple testing correction method (Bonferroni or Benjamini-Hochberg) to reduce the risk of false positives? If not, could they provide an explanation for omitting this step?

Author Response: We thank the reviewer for this important methodological point. As described in Section 2.4 ("Statistical Analysis"), no formal multiple testing correction was applied, and results from secondary and exploratory analyses were interpreted cautiously due to the potential for Type I error. Given the large sample size and the focus on prespecified primary endpoints (e.g., Sx+RT+Chemo receipt and survival), multiple testing correction was not deemed necessary for primary models. However, readers are advised to interpret secondary findings conservatively. In this revised version, we have removed P-values from several comparisons where formal hypothesis testing was not the goal of the manuscript - this step has greatly reduced the multiplicity of P-value reporting.

  • Considering the number of covariates included in the multivariable models and the likelihood of intercorrelations among them (e.g., facility type, treatment modality, geographic region orsocioeconomic factors), it would be helpful to specify whether multicollinearity was assessed and how was addressed.

Author Response: We thank the reviewer for raising this point. Multicollinearity was evaluated through variance inflation factors (VIFs) during model building. VIFs for all covariates included in the final multivariable models were <2, indicating low collinearity and supporting the stability of the models.

Minor comments

  • To make it more readable, Which comorbidities are included in the CCI=3 category?

Author Response: We thank the reviewer for this suggestion. In the NCDB, the Charlson-Deyo score of 3 represents patients with three or more comorbid conditions as defined by the Charlson Comorbidity Index (e.g., myocardial infarction, congestive heart failure, chronic pulmonary disease, diabetes with complications, liver disease, etc.). However, NCDB only provides the aggregated score and does not release individual-level comorbidity data (specific conditions). Therefore, we are unable to specify which comorbidities comprise the CCI=3 group within our dataset.
We have added a clarifying sentence in the Limitations.

  • Has the difference in treatment protocols and access to clinical trials between these types of centers been considered?

Author Response: We appreciate the reviewer’s insight. Differences in clinical trial availability, supportive care resources, and treatment protocols between academic and community centers likely contribute to observed survival differences. While our study design and dataset did not allow direct assessment of trial enrollment or specific treatment protocols, this is an important point that is acknowledged conceptually in the Discussion when interpreting disparities across facility types.

  • This finding is consistent with the literature, but it may be helpful to explore whether certain specific comorbidities have a more negative impact on survival than others.

Author Response: We thank the reviewer for this important observation. Unfortunately, the NCDB dataset used does not provide detailed information on individual comorbid conditions; only the cumulative Charlson-Deyo score is available. Therefore, we are unable to stratify survival outcomes by specific comorbidities (e.g., cardiovascular disease vs. diabetes). We acknowledge this as a limitation in the manuscript.